# Understanding and responding to COVID-19 in Wales: protocol for a privacy-protecting data platform for enhanced epidemiology and evaluation of interventions

Jane Lyons [iD],[1] Ashley Akbari [iD],[1] Fatemeh Torabi,[1] Gareth I Davies,[1] Laura North,[1] Rowena Griffiths,[1] Rowena Bailey,[1] Joseph Hollinghurst [iD],[1] Richard Fry [iD],[1] Samantha L Turner,[1] Daniel Thompson,[1] James Rafferty,[1] Amy Mizen,[1] Chris Orton [iD],[1] Simon Thompson,[1] Lee Au-Yeung,[1] Lynsey Cross,[1] Mike B Gravenor,[2] Sinead Brophy,[1] Biagio Lucini,[1] Ann John [iD],[1] Tamas Szakmany,[3,4] Jan Davies,[5] Chris Davies,[5] Daniel Rh Thomas,[6] Christopher Williams,[6] Chris Emmerson,[6] Simon Cottrell,[6] Thomas R Connor,[7] Chris Taylor,[8] Richard J Pugh,[9] Peter Diggle,[10,11] Gareth John,[12] Simon Scourfield,[12] Joe Hunt,[12] Anne M Cunningham,[12] Kathryn Helliwell,[13] Ronan Lyons [iD] [1]

For numbered affiliations see end of article.

**Correspondence to**
Jane Lyons;
j.lyons@swansea.ac.uk

## ABSTRACT

**Introduction** The emergence of the novel respiratory SARS-CoV-2 and subsequent COVID-19 pandemic have required rapid assimilation of population-level data to understand and control the spread of infection in the general and vulnerable populations. Rapid analyses are needed to inform policy development and target interventions to at-risk groups to prevent serious health outcomes. We aim to provide an accessible research platform to determine demographic, socioeconomic and clinical risk factors for infection, morbidity and mortality of COVID-19, to measure the impact of COVID-19 on healthcare utilisation and long-term health, and to enable the evaluation of natural experiments of policy interventions.

**Methods and analysis** Two privacy-protecting population-level cohorts have been created and derived from multisourced demographic and healthcare data. The C20 cohort consists of 3.2 million people in Wales on the 1 January 2020 with follow-up until 31 May 2020. The complete cohort dataset will be updated monthly with some individual datasets available daily. The C16 cohort consists of 3 million people in Wales on the 1 January 2016 with follow-up to 31 December 2019. C16 is designed as a counterfactual cohort to provide contextual comparative population data on disease, health service utilisation and mortality. Study outcomes will: (a) characterise the epidemiology of COVID-19, (b) assess socioeconomic and demographic influences on infection and outcomes, (c) measure the impact of COVID-19 on short-term and longer-term population outcomes and (d) undertake studies on the transmission and spatial spread of infection.

**Ethics and dissemination** The Secure Anonymised Information Linkage-independent Information Governance

### Strengths and limitations of this study

► Rapid access to multiple data sources on a complete population.
► Great variety of individual and household-level data on demography, disease status, morbidity, mortality and viral genomics to support a wide range of studies on the evolution of the epidemic in Wales.
► Ability to support hierarchical analyses at varying geographical units: private residences, care homes, educational settings and healthcare facilities to examine spatial spread and transmission of SARS-CoV-2 to inform and evaluate targeting of interventions.
► However, routine data do not capture data on some important aspects, such as quality of life.

Review Panel has approved this study. The study findings will be presented to policy groups, public meetings, national and international conferences, and published in peer-reviewed journals.

## INTRODUCTION

Understanding and controlling the COVID-19 pandemic is a rapidly changing, complex issue that requires near real-time local data, analyses, modelling and multidisciplinary team science to devise, implement and evaluate a wide variety of intersectoral and cross-sectoral interventions to minimise population harm.[1]

As the pandemic evolves, a wide range of issues need to be considered including the spread of infection in the general and vulnerable populations; health service resilience; indirect harm minimisation; and effectiveness of control policies and interventions.

Responding to this challenge, the Welsh Government created a COVID-19 Technical Advisory Group (TAG) to provide rapid assimilation of available evidence and guide analysis of data to inform policy development and appraisal. Insight from linked data is seen as being essential to understand the evolving epidemic. TAG commissioned the support of analyses conducted through the Secure Anonymised Information Linkage (SAIL) Databank (www.saildatabank.com) to formulate evidence and advice to underpin its work in responding to COVID-19.[2–5] SAIL is a state of the art, remotely accessible, privacy-protecting system, accredited under the Digital Economy Act. SAIL holds and provides access to linked de-identified data from multiple sources at individual, household and multiple ecological levels, for the population of Wales. The SAIL Databank has previously supported numerous types of clinical and population studies, including cohorts, evaluations of natural experiments and embedded trials.[6–13]

This paper describes the development of two population-based cohorts in Wales, derived from multiple data sources to provide near real-time, in-pandemic intelligence and analytics to TAG in relation to the following broad objectives:

### Primary objectives

1. Determine demographic, socioeconomic and clinical risk factors for infection, morbidity and mortality-related to COVID-19.
2. Determine the risk of COVID-19 infection and outcomes in occupational groups.
3. Measure the population impact of COVID-19 on healthcare utilisation.

### Secondary objectives

1. Create a platform to enable the evaluation of policies and interventions aimed at controlling the epidemic, whether clinical or non-pharmaceutical in nature.
2. Provide access to these derived population-based cohorts and linked data sources to organisation and people with relevant skills and expertise within the National Health Service (NHS), academia and government.

## METHODS

### Study design and population

The cohorts were derived from de-identified linked data from the SAIL Databank. We created two population-based cohorts derived from multiple demographic and healthcare data sources (figure 1).

▶ The C20 cohort consists of all people alive and known to the NHS in Wales from the 1 January 2020 with follow-up until 31 May 2020. We include people who moved into or were born in Wales after 1 January 2020. Follow-up data will be added prospectively and the C20 cohort will be updated on a monthly basis in line with a full month of coverage of available data. Linkage to other data sources is also available beyond

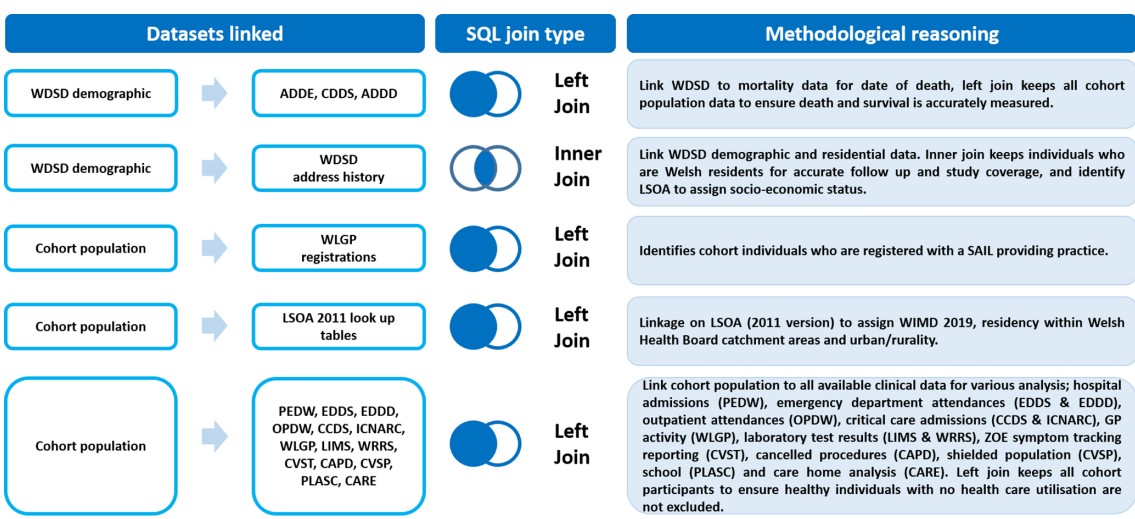

**Figure 1** Data linkage of multiple demographic and healthcare data sources used in the creation of two population-wide cohorts: C20 and C16. ADDD, Annual District Death Daily; ADDE, Annual District Death Extract; CAPD, Postponed Admitted Procedures; CARE, Care homes Index; CCDS, Critical Care Data Set; CDDS, Consolidated Death Data Source; CVSP, COVID-19 Shielded People list; CVST, KCL Zoe Symptom Tracker App; EDDD, Emergency Department Dataset Daily; EDDS, Emergency Department Data Set; ICNARC, Intensive Care National Audit & Research Centre; LIMS, Laboratory Information Management System; LSOA, Lower Layer Super Output Area; OPDW, Out Patient Dataset for Wales; PEDW, Patient Episode; PLASC, Pupil Level Annual School Census; SAIL, Secure Anonymised Information Linkage; SQL, Structured Query Language; WDSD, Wash Demographic Service Dataset; WIMD, Welsh Index of Multiple Deprivation; WLGP, Welsh Longitudinal General Practise; WRRS, Wales Results Reporting Service.

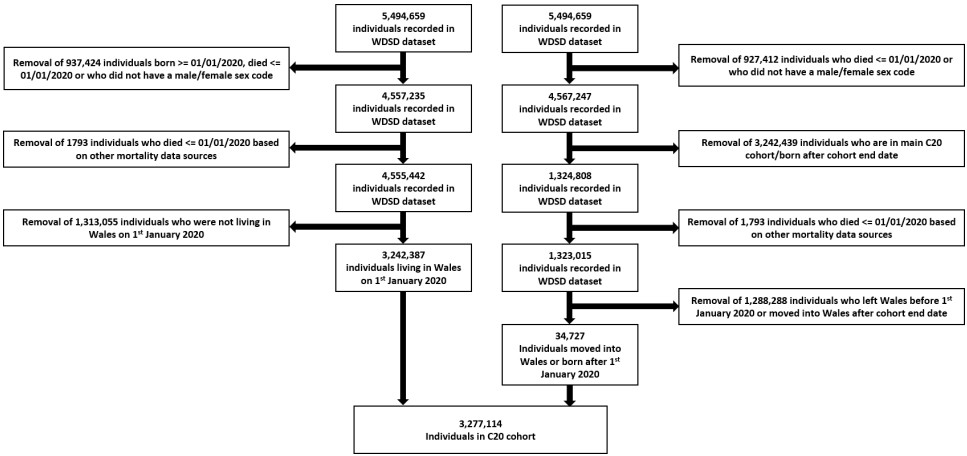

**Figure 2** Flow diagram of the C20 cohort inclusion criteria. WDSD, Wales Demographic Service Dataset.

the cohort end date where the frequency and quality of each data source allow its use. Some datasets are analysed daily.

► The C16 cohort includes all individuals living in Wales and known to the NHS on the 1 January 2016 with follow-up to 31 December 2019. C16 is designed to provide counterfactual and contextual comparative data on population health service utilisation and mortality rates.

Membership of both cohorts is based on the inclusion of a person's residence in Wales, registered to a Welsh General Practice (GP), a free to use NHS system at the point of primary care registration in the UK (figure 2). This is recorded within the Wales Demographic Service Dataset (WDSD). People are censored by study endpoints or migration out of Wales.

### Data sources

Baseline populations are created using the weekly updated WDSD, the monthly updated Office for National Statistics (ONS) mortality registry data known as the Annual District Death Extract, two new COVID-19 daily data sources: the Consolidated Death Data Source created by NHS Wales Informatics Service (NWIS) and the Annual District Death Daily from ONS.

### Anonymised linkage fields

Linkage fields are used to anonymously link between data sources in the SAIL Databank. SAIL uses a multiple encryption system in which a trusted third party, the NWIS, uniquely matches identities to an Anonymised Linkage Field (ALF) and residences to a Residential Anonymised Linkage Field (RALF) before uploading data to SAIL.[2 3 14]

### Demographic data

The cohorts include the following variables: ALF, age, sex, date of death, date of movement out of Wales, RALF and Care Home Anonymised Linkage Fields (CHALFs) for older people at cohort inception. The CHALF was derived from a data extract from Care Inspectorate Wales in 2020 for all adult care home settings.[8] Geographical variables associated with the RALF and CHALF include

Lower layer Super Output Area (LSOA) 2011 boundaries that are small statistical areas containing around 1500 people. LSOA 2011 has been mapped to the Welsh Index of Multiple Deprivation (WIMD) version 2019 to derive deprivation quintiles; Welsh health board of residence; and urban/rurality categories.[15 16] Using the Welsh Government's Pupil-Level Annual School Census (PLASC), the school population can also be linked to the cohorts for analyses by the school network.[17]

In addition, permission has been granted to embed occupation and role categories from electronic staff records of all NHS health boards and trusts, local authority social care workers and education staff. For healthcare workers, the electronic staff records system used in all health boards and trusts (111 000) are categorised by whether roles involve direct patient care or not and by occupational groups: Additional Professional Scientific and Technical; Additional Clinical Service; Administrative and Clerical; Allied Health Professionals; Estates and Ancillary; Healthcare Scientist; Medical and Dental, Nursing and Midwifery Registered; and Students. Social care workers are registered (https://socialcare. wales/) and grouped into social workers, child home workers and domiciliary care workers (estimated 32 000). Educational staff records (estimated 70 000) include categories for teachers, support and administrative staff. This information is collected from the annual School Workforce Annual Census held by the Welsh Government.[18] Data on care home staff are collected as part of Public Health Wales (PHW) testing of all staff and residents and made available in SAIL through the standard Laboratory Information Management System dataset. Permission has also been granted to link 2011 ONS census fields on ethnicity, occupation, housing tenure, over-crowding and socioeconomic status (SES). Ethnicity codes are derived from multiple health and social data sources mapped to Census 2011 groupings.[19]

### Health data

All hospital admissions, outpatient and emergency department attendances treated in NHS hospitals as well

as GP data on all diagnoses and treatments from SAIL providing practices (80% population coverage) are available for cohort participants.[20] As of the beginning of 2020, we have added:

▶ Daily GP respiratory and COVID-19 codes for 100% of Welsh GP practices.
▶ Daily COVID-19 antigen test results.
▶ Bi-weekly data on participants reporting symptoms through the King's College London/ZOE symptom tracking app.[21]
▶ Weekly critical care data from the Intensive Care National Audit and Research Centre.[22 23]
▶ Bi-weekly COVID-19 viral genomic variant call format data and viral lineage assignments from PHW.[24]
▶ Monthly community dispensing data from pharmacies providing NHS issued prescriptions, backdated to 2016.[25]

### Exposure variables and potential confounding factors

A number of exposure variables will be used to contextualise the study primary outcomes, including age, sex, SES and clinical risk groups.

SES will be derived from WIMD with quintile 1 including the 20% most deprived areas, and also at individual/household level from the 2011 Census using the following codes: approximated social grade (SCGPUK11), highest level of qualification (HLQPUK11) and National Statistics Socioeconomic Classification.[26]

Clinical risk groups have been derived from those used in scientific papers:

▶ Predictors of influenza and COVID-19 outcomes.[27 28]
▶ Published phenotype (disease conditions) libraries including the 308 phenotypes created by the Cardiovascular disease research using linked bespoke studies and electronic health records (CALIBER) study.[29]
▶ Commonly used comorbidity indices (Charlson and Elixhauser).[30 31]
▶ Frailty indices (electronic Frailty Index for GP data and Hospital Frailty Risk Score).[32–34]
In order to compare and combine results with other studies, we will replicate the 19 clinical groups included in a similar study in Scotland.[35]

Microbiological testing data will be de-duplicated and used to generate case-data for standard case definitions, agreed at UK level where possible.

Body mass index will be categorised as <20, 20–24, 25–29, 30–39, ≥40 kg/m$^2$; and smoking status categorised into four groups: current smoker, non-smoker, ex-smoker and not recorded for patients with no data on smoking, replicated from a study carried out in Scotland.[35]

### Statistical analysis

We will describe baseline characteristics for exposures and outcomes of interest using means, medians, proportions, ORs and rate ratios with appropriate measures of dispersion. We will report on the prevalence of missing data by variable and use two-tailed hypothesis tests with a 5% significance level.

Non-independence of observations measured over time or within associated clusters, for example, GP or households will be taken into account using random effects. We will use causal frameworks where causal relationships are implied.[36] Hypotheses being tested will be stated in advance and plans will be drawn up for each research project. Analyses will primarily be conducted in the R statistical programming language.[37 38]

### Analyses

We will test associations for demographic, socioeconomic and clinical risk factors for COVID-19 infection and associated morbidity and mortality. COVID-19 infection will be defined in a number of ways: (a) positive SARS-CoV-2 laboratory antigen test; (b) clinical diagnosis of COVID-19 infection in GP records, intensive care, or hospital discharge records; (c) ONS mortality records listing COVID-19 as the underlying or contributory cause; and (d) positive serology report (when available).

Planned analyses include:

▶ Incidence of COVID-19 over time and by geography and demographic groups.
▶ Influence of area deprivation and individual SES metrics on infection and outcomes.
▶ Impact of COVID-19 on short-term (<6 months) and longer-term population outcomes such as changes in health service utilisation and excess, overall and disease-specific mortality.
▶ Description of the extent of clustering of cases within all available residential, educational, occupational and geographic units, thus providing signatures of spatial spread at defined levels, and between levels, thought to have played a crucial role in transmission.

We will investigate the relationship between health (physical and mental), socioeconomic and environmental factors, such as self-rated health, limiting long-term illness, housing tenure, over-crowding, education status, and occupation on infection risk and outcome.

Changes in healthcare utilisation will be assessed by measuring differences preinfection and postinfection for: NHS111 telephone calls, GP consultations, emergency department attendances, hospital admissions and length of stay, and intensive care admissions.

Analytical techniques will include descriptive statistics, univariate and multivariate generalised linear mixed models, survival analyses and the use of self-controlled case series for temporary risk factors.[39] Relationships between variables will be clarified before specific analyses.

### Cohort characteristics

The C16 and C20 cohorts have been constructed from patients registered with all GPs in Wales (table 1).

The power to detect relevant outcomes will be assessed as the pandemic evolves. There are plans to collaborate with researchers across the UK and collating data from similar cohorts,[35] to maximise power, support the evaluation of natural experiments in policy and its timing around disease control, and exit strategies from the

**Table 1** C16 and C20 cohort demographics till the end of May 2020

| Cohort | C16 | C20 |
|---|---|---|
| Individuals (N) | 3 087 032 | 3 277 114 |
| Cohort start date | 1 January 2016 | 1 January 2020 |
| Cohort end date | 31 December 2019 | 31 May 2020 |
| Deaths in period | 117 565 (3.8%) | 16 380 (0.5%) |
| Full coverage (cohort end date=31 December 2019/31 May 2020) | 2 651 957 (85.9%) | 3 237 389 (98.8%) |
| Registered with a SAIL providing practice (registration end date >cohort start date) | 2 608 761 (84.5%) | 2 666 331 (81.4%) |
| Mean age (SD) | 41.3 (23.7) | 41.9 (23.8) |
| Sex | | |
| Female | 50.1% | 50.1% |
| WIMD 2019 quintile* | | |
| 1 | 20.3% | 19.1% |
| 2 | 19.9% | 18.5% |
| 3 | 20.1% | 18.4% |
| 4 | 19.7% | 18.1% |
| 5 | 19.9% | 18.3% |
| Missing WIMD | 0.0% | 7.7% |

*WIMD 2019 quintile: 1=most deprived, 5=least deprived, please note a one decimal place rounding error.
SAIL, Secure Anonymised Information Linkage; WIMD, Welsh Index of Multiple Deprivation.

lockdown on physical restrictions that commenced on 23 March 2020.

The individual datasets that comprise the cohorts are held on the globally accessible SAIL Databank available to accredited researchers.

### Proposed future developments

As the pandemic evolves so will policies and practices to control the epidemic and mitigate negative consequences. As these develop, we plan to use the cohorts as a platform for their evaluation, by linking dates and presence of interventions as data become available. There are subtle differences in the timing and approaches to controlling the epidemic in diverse settings and in exiting lockdown across the four UK nations. This provides opportunities for collaborative and timely evaluation of natural experiments of policies and approaches across the UK, which would refine evidence-based exit strategies. We are also keen to contribute to international initiatives.

### Patient and public involvement

This study is based on an extension of the developing Wales Multimorbidity Cohort (WMC). CD and JD are members of the public who were involved in the design of the WMC and C20/C16 studies. Additional members of the public are in the process of being recruited to the research steering committee to represent the views of health, social care and educational staff.

### ETHICS AND DISSEMINATION

SAIL's independent Information Governance Review Panel (IGRP)[5] has approved a submission to allow the use of WMC with additional data flows to aid the COVID-19 research response (SAIL project 0911). IGRP applications are scrutinised by members of the public; only those applications that can demonstrate privacy protection and are in the public interest are approved. SAIL's Consumer Panel, comprising members of the public, were consulted during the development of WMC. Two members of the public were recruited to the study steering group following approval.

**Author affiliations**
[1]Population Data Science, Swansea University Medical School, Swansea, UK
[2]Institute of Life Sciences, Swansea University Medical School, Swansea, UK
[3]Department of Anaesthesia, Intensive Care and Pain Medicine, Division of Population Medicine, Cardiff University, Cardiff, UK
[4]Aneurin Bevan University Health Board, Newport, UK
[5]Members of the public, Swansea, UK
[6]Public Health Wales NHS Trust, Cardiff, UK
[7]School of Biosciences, Cardiff University, Cardiff, UK
[8]School of Social Sciences, Cardiff University, Cardiff, UK
[9]Glan Clwyd Hospital, Betsi Cadwaladr University Health Board, Rhyl, UK
[10]Faculty of Health and Medicine, Lancaster University, Lancaster, UK
[11]Epidemiology and Population Health, University of Liverpool, Liverpool, UK
[12]NHS Wales Informatics Service, Cardiff, UK
[13]Welsh Government, Cardiff, UK

**Acknowledgements** The authors would like to acknowledge that this work uses data provided by patients and collected by the National Health Service (NHS) as part of their care and support and the Understanding Patient Data initiative. They

would also like to acknowledge all data providers who make anonymised data available for research. They wish to acknowledge the collaborative partnership that enabled acquisition and access to the de-identified data, which led to this output. The collaboration was led by the Swansea University Health Data Research UK team under the direction of the Welsh Government Technical Advisory Group. The team includes the following groups and organisations: the Secure Anonymised Information Linkage (SAIL) Databank, Administrative Data Research Wales, NHS Wales Informatics Service, Public Health Wales, NHS Shared Services and the Welsh Ambulance Service Trust. All research conducted has been completed under the permission and approval of the SAIL independent Information Governance Review Panel project number 0911.

**Contributors** All authors contributed to the conception and or design of aspects of the study. JL is the lead analyst for the Wales Multimorbidity Cohort creation and designed the data framework for the C20/C16 cohorts. JL, AA, FT, GID, LN, RG, RB, JH, RF, SLT, DT, JR, AM, CO, ST, LA-Y, TS, DRT, CE, TRC, CT, RJP, GJ, SS, JH, AMC created meta-data, prepared or linked data sources to create the cohort. JL, AA, FT, GID, LN, RG, RB, JH, RF, SLT, DT, JR, AM, CO, ST, LA-Y, LC, MBG, SB, BL, AJ, TS, JD, CD, DRT, CW, CE, SC, TRC, CT, RJP, PD, GJ, SS, JH, AMC, KH and RL contributed to the drafting of the manuscript and gave final approval of the version to be published. RL is the principle investigator and guarantor of the study.

**Funding** The WMC, on which components of this study are based, is funded by Health Data Research UK (HDR-9006) and the Medical Research Council (MR/S027750/1). Funding for the COVID-19 extension is through the Medical Research Council (MR/V028367/1). Health Data Research UK is funded by: UK Medical Research Council; Engineering and Physical Sciences Research Council; Economic and Social Research Council; National Institute for Health Research (England); Chief Scientist Office of the Scottish Government Health and Social Care Directorates; Health and Social Care Research and Development Division (Welsh Government); Public Health Agency (Northern Ireland); British Heart Foundation; and Wellcome.

**Disclaimer** The views and opinions expressed therein are those of the authors and do not necessarily reflect those of the funding agencies, NHS organisations or Welsh Government.

**Competing interests** None declared.

**Patient and public involvement** Patients and/or the public were involved in the design, or conduct, or reporting, or dissemination plans of this research.

**Patient consent for publication** Not required.

**Provenance and peer review** Not commissioned; externally peer reviewed.

**Open access** This is an open access article distributed in accordance with the Creative Commons Attribution 4.0 Unported (CC BY 4.0) license, which permits others to copy, redistribute, remix, transform and build upon this work for any purpose, provided the original work is properly cited, a link to the licence is given, and indication of whether changes were made. See: https://creativecommons.org/licenses/by/4.0/.

**ORCID iDs**
Jane Lyons http://orcid.org/0000-0002-4407-770X
Ashley Akbari http://orcid.org/0000-0003-0814-0801
Joseph Hollinghurst http://orcid.org/0000-0002-3556-2017
Richard Fry http://orcid.org/0000-0002-7968-6679
Chris Orton http://orcid.org/0000-0002-9561-2493
Ann John http://orcid.org/0000-0002-5657-6995
Ronan Lyons http://orcid.org/0000-0001-5225-000X

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
