## [Reviewer comments · BMJ Open]

ARTICLE DETAILS

TITLE (PROVISIONAL)	Understanding and responding to COVID-19 in Wales: protocol for a privacy protecting data platform for enhanced epidemiology and evaluation of interventions.
AUTHORS	Lyons, Jane; Akbari, Ashley; Torabi, Fatemeh; Davies, Gareth; North, Laura; Griffiths, Rowena; Bailey, Rowena; Hollinghurst, Joseph; Fry, Richard; Turner, Samantha L.; Thompson, Daniel; Rafferty, James; Mizen, Amy; Orton, Chris; Thompson, Simon; Au-Yeung, Lee; Cross, Lynsey; Gravenor, Mike; Brophy, Sinead; Lucini, Biagio; John, Ann; Szakmany, Tamas; Davies, Jan; Davies, Chris; Thomas, Daniel; Williams, Christopher; Emmerson, Chris; Cottrell, Simon; Connor, Thomas; Taylor, Chris; Pugh, Richard; Diggle, Peter; John, Gareth; Scourfield, Simon; Hunt, Joe; Cunningham, Anne Marie; Helliwell, Kathryn; Lyons, Ronan

VERSION 1 – REVIEW

REVIEWER	Hassan Harb Lebanese University, Lebanon
REVIEW RETURNED	06-Aug-2020

GENERAL COMMENTS	In this paper, the authors propose a protocol for data communication and analysis responding to COVID-19 in Wales. Unfortunately, the paper suffers from several drawbacks: 1- The used English language is very poor. Many sentences are very long and not comprehensible.2- The organization is a weakness point in this paper. The whole structure must be revised.3- The scientific models are missed. The paper is dedicated to an application more than to a research article.4- I think that the paper does not fit a journal format and it could be more suitable to publish as report.5- The obtained results are very superficial and not convincing.
--

REVIEWER	An Zhang The Second Affiliated Hospital of Chongqing Medical University. Chongqing, China.
REVIEW RETURNED	18-Aug-2020

GENERAL COMMENTS	The primary objectives of this study are to determine demographic, socioeconomic and clinical risk factors for infection, morbidity, and mortality related COVID-19. However, I did not see any potential risk factors or predictors for morbidity or mortality in this protocol, such as older age, high SOFA score, or high d-dimer value (Fei Zhou et al. Lancet. 2020 Mar 28;395(10229):1054-1062). Can you list the potential risk factors you would like to analyze?
---

REVIEWER	Qiongjing Yuan Xiangya Hospital, Central South University, China
REVIEW RETURNED	05-Sep-2020

GENERAL COMMENTS	Generally speaking, the authors made a protocol for a privacy protecting data platform for enhanced epidemiology and evaluation of interventions which will be useful. Before acceptance, it is better to include a flow chart to summary the whole protocol.
---

VERSION 1 – AUTHOR RESPONSE

Reviewers' Comments to Author:

Reviewer: 1

Reviewer Name: Hassan Harb

Institution and Country: Lebanese University, Lebanon

Please state any competing interests or state 'None declared': None declared

In this paper, the authors propose a protocol for data communication and analysis responding to COVID-19 in Wales. Unfortunately, the paper suffers from several drawbacks:

1- The used English language is very poor. Many sentences are very long and not comprehensible.

Thank you. We appreciate that some of the sentences are quite long and structured. This is a style issue which has both pros and cons. Following your review we have gone through the paper and simplified the structure of sentences, making them shorter.

2- The organization is a weakness point in this paper. The whole structure must be revised.

We were surprised to read this comment. Our paper is a protocol paper not a results paper. We followed the journal guidance and modelled the paper on a very similar paper published recently by BMJ Open which describes the development of a similar protocol in Scotland – reference 35 in our paper. <https://bmjopen.bmj.com/content/bmjopen/10/6/e039097.full.pdf>

3- The scientific models are missed. The paper is dedicated to an application more than to a research article.

Our manuscript is a protocol paper; it describes the development of a set of linked data on the population of Wales (UK) to study the unfolding pandemic. It will be used to investigate the impact of CoVID-19 on the population and enable the evaluation of policies and interventions. You are correct in describing it as an application and not a research article. It will form the basis of many research articles. We do not consider that scientific models should be included in such an overarching protocol. These will be specified for each succeeding paper that addresses a scientific question.

4- I think that the paper does not fit a journal format and it could be more suitable to publish as report.

We have followed the journal guidance on protocol papers and modelled our paper on a very similar paper from Scotland recently published by BMJ Open.

5- The obtained results are very superficial and not convincing.

The results merely show the number of people within the cohort and coverage within datasets. We consider that this demonstrates the ability to construct such a multi-sourced cohort.

Reviewer: 2

Reviewer Name: An Zhang

Institution and Country: The Second Affiliated Hospital of Chongqing Medical University. Chongqing, China.

Please state any competing interests or state 'None declared': None declared

The primary objectives of this study are to determine demographic, socioeconomic and clinical risk factors for infection, morbidity, and mortality related COVID-19. However, I did not see any potential risk factors or predictors for morbidity or mortality in this protocol, such as older age, high SOFA score, or high d-dimer value (Fei Zhou et al. Lancet. 2020 Mar 28;395(10229):1054-1062). Can you list the potential risk factors you would like to analyze?

The risk factors we would like to examine are described in the paper, albeit it in a number of places under each dataset. We mention that we would determine demographic, socioeconomic and clinical risk factors for infection and outcomes. Factors described in the paper include age group, sex and area deprivation score, occupational status (health care, social care and educational workers), ethnicity, housing status and clinical risk groups. We have copied below paragraphs that refer to these variables.

“In addition, permission has been granted to embed occupation and role categories from electronic staff records of all NHS health boards and trusts, local authority social care workers and education staff. For healthcare workers, the electronic staff records system used in all health boards and trusts (111,000) are categorised by whether roles involve direct patient care or not and by occupational groups: Additional Professional Scientific and Technical; Additional Clinical Service; Administrative and Clerical; Allied Health Professionals; Estates and Ancillary; Healthcare Scientist; Medical and Dental, Nursing and Midwifery Registered; and Students. Social care workers are registered (<https://socialcare.wales/>) and grouped into social workers, child home workers and domiciliary care workers (estimated 32,000). Educational staff records (estimated 70,000) include categories for teachers, support and administrative staff. This information is collected from the annual School Workforce Annual Census (SWAC) held by Welsh Government.[18] Data on care home staff is collected as part of Public Health Wales testing of all staff and residents and made available in SAIL through the standard Laboratory Information Management System (LIMS) dataset. Permission has also been granted to link 2011 ONS census fields on ethnicity, occupation, housing tenure, over-crowding and socio-economic status. Ethnicity codes are derived from multiple health and social data sources mapped to Census 2011 groupings.[19]”

“Socio-economic status (SES) will be derived from WIMD with quintile 1 including the 20% most deprived areas, and also at individual/household level from 2011 census using the following codes: approximated social grade (SCGPUK11); highest level of qualification (HLQPUK11), and National Statistics Socio-economic Classification (NSSEC).[26]”

“Clinical risk groups have been derived from those used in scientific papers:

- Predictors of influenza and COVID-19 outcomes.[27-28]
- Published phenotype (disease conditions) libraries including the 308 phenotypes created by the CALIBER study.[29]

- Commonly used comorbidity indices (Charlson and Elixhauser).[30-31]
- Frailty indices (electronic Frailty Index for GP data and Hospital Frailty Risk Score).[32-34]

In order to compare and combine results with other studies we will replicate the 19 clinical groups included in a similar study in Scotland.[35]”

It is worth noting, that early risk-prediction models developed from relatively limited datasets from the early phase of the pandemic are likely to suffer from significant issues in terms of transferability to other healthcare systems and their fragility due to lack of external validation and calibration has been highlighted. We believe, that our large, population-based dataset using routinely collected clinical parameters could be used for building more transferable and well calibrated risk prediction tools.

SOFA scores, although they are ubiquitously used in critical care, are not at this moment available in an easily collected digital format in the Welsh NHS. We have access to this data at the point of ICU admission in our patients, however we don't have means of collecting this information longitudinally. Nor do we have access to d-dimer values at this point. We are currently in discussions with colleagues in NHS Wales Informatics Service to investigate the possibility of acquiring such laboratory data as we have done in the past.

Reviewer: 3

Reviewer Name: Qiongjing Yuan

Institution and Country: Xiangya Hospital, Central South University, China

Please state any competing interests or state 'None declared': None declared

Generally speaking, the authors made a protocol for a privacy protecting data platform for enhanced epidemiology and evaluation of interventions which will be useful. Before acceptance, it is better to include a flow chart to summary the whole protocol.

Thank you for your comments. We have created and included a flow chart as requested.